# The relationship between family functioning and depression among adolescents in China during the normalization stage of the COVID-19 epidemic: The mediating role of resilience

Yameng Feng[1ᴼ], Yang Zhou[2,3ᴼ], Wenzhen Li[4], Qingzhou Cheng[1], Cen Gao[1], Shu Yan[2], Na Lv[2], Yaofei Xie[2], Taimin Wu[2], Rong Nie[1], Hongping Zhang[1], Dajie Chen[1]*

1 Department of Health Services and Management, College of Medicine and health Science, Wuhan Polytechnic University, Wuhan, Hubei, China, 2 Department of Psychiatry, Wuhan Mental Health Center, Wuhan, Hubei, China, 3 Office of Psychosocial Services, Wuhan Hospital for Psychotherapy, Wuhan, Hubei, China, 4 JC School of Public Health and Primary Care, The Chinese University of Hong Kong, Hong Kong Special Administrative Region of China, China

ᴼ These authors contributed equally to this work.
* 22113061@whpu.edu.cn

## Abstract

### Background

In May 2020, China shifted its COVID-19 pandemic prevention and control status from "emergency" to "regularization". However, thus far, limited research has investigated the mental health of adolescents in Wuhan during this phase. This study examined the mental health status of adolescents in Wuhan during the regular prevention and control against COVID-19 period and explored whether psychological resilience mediated the relationship between adolescent family functioning and depression.

### Methods

A cross - sectional survey was conducted from September to December 2022 in Wuhan. Using a census methodology, 2,410 students from a secondary vocational school were recruited for the study. Multivariate logistic regression was performed to investigate the independent effect of family functioning and psychological resilience on adolescent depression. Structural equation modeling was used to analyze the mediating role of psychological resilience in the relationship between family functioning and depression.

### Results

The detection rate of depression among adolescent students was 35.77%. Both family functioning and psychological resilience were significantly and negatively associated with the prevalence of depression, with OR=0.800 (95%CI: 0.772–0.830)

**Data availability statement:** Participant confidentiality restrictions prohibit the authors from making the data set publicly available. The study was approved by the Ethics Committee of Wuhan Mental Health Center (KY2022.11.01). Any queries about the data may be directed to ethics committee secretary, Yanqin Xu, Email: xuyanqin8327@dingtalk.com

**Funding:** National Center for Mental Health, China Education Development Foundation, Center for Student services and Development, ministry of education, P.R. China, "Construction and Evaluation of a Collaborative Model for Mental Health Services in Education for Primary and Secondary School Students" (XS24A035) the funders had no role in study design, data collection and analysis, decision to publish, or preparation of the manuscript.

**Competing interests:** The authors have declared that no competing interests exist.

and OR=0.950 (95%CI: 0.938–0.962) respectively. Family functioning was not only directly related to depression ($\beta$=-0.575 [95% CI: -0.655, -0.505]), but also through the mediating effect of psychological resilience ($\beta$=-0.135 [95% CI: -0.173, -0.099]). Psychological resilience partially mediated the effect of family functioning on depression, with the mediating effect accounting for 19.72% of the total effect.

## Conclusions

The COVID-19 pandemic has substantially exacerbated mental health issues among adolescents in Wuhan, China and further attention is required. The detrimental impact of poor family functioning on adolescents' depression may be reduced by fostering psychological resilience.

## Introduction

A World Health Organization report documented that depression is the leading cause of illness and disability in children and adolescents aged 10–19 years globally and the third most common cause of disability-adjusted life-years among adolescents [1,2]. Adolescent depression can lead to poor academic performance and impaired social functioning and is a high-risk factor for substance use disorders (e.g., smoking and drinking) and suicidal behaviors [3]. Unlike adult depression, the harm caused by adolescent depression not only occurs in the present, its far-reaching effects may also persist through adulthood. Some aspects of adolescent depression may also be viewed as subordinate forms of early episodes of adult depression and are closely related to later recurrence [4]. Non-clinical depression in adolescents has been shown to be the most important cause of adult depression, with 67% of cases potentially considered clinical depression [5].

High school is an important stage of adolescent growth during which adolescents bear heavy learning tasks and face high pressure caused by college entrance examinations and stress from interpersonal relationships, making it a period with a high incidence of adolescent depression. A large-sample study from a mental health literacy survey of adolescents aged 10–19 in 29 provinces in China determined that the prevalence of depression increased with grade level, with a detection rate of 10.0%, 30%, and as high as 40.0% in elementary, junior high, and senior high school, respectively [6]. In December 2019, the COVID-19 pandemic unexpectedly swept across the globe as a major public health event. Numerous studies have confirmed that the interaction between the environmental characteristics of the pandemic, (e.g., suddenness and hazards) and adolescent immaturity led to young people experiencing more depression and anxiety issues [7–9]. However, most studies on mental health issues during the pandemic focused on the outbreak phase. In May 2020, China's pandemic prevention and control moved from the "emergency" to the "regularization" stage; that is, a phase of regular prevention and control in which one must be on guard against the threat of an epidemic recurrence while also adapting to the aftershocks of the epidemic such as the negative impacts of recession and market

downturns on people's lives, education, and work. However, to date, both in China and abroad, relatively limited research has investigated adolescents' mental health when facing the complexity and uncertainty of the present and the future; in particular, whether the mental health of adolescents in Wuhan during the eye of the pandemic was more serious than that during the "state of emergency" pandemic stage. Therefore, more research should be conducted to detect the epidemiological characteristics of depression among adolescents in Wuhan during the normalization stage of the COVID-19 epidemic and to explore the preventive approaches to adolescent depression.

Several studies have examined how to promote adolescents' healthy development through the framework of developmental assets [10,11]. Such assets are relevant experiences, relationships, skills, and values that promote healthy developmental outcomes for adolescents, including both external (e.g., family functioning) and internal (e.g., psychological resilience) resources [11]. Family functioning is a comprehensive assessment of the family, including family members' relationships and adaptability, the family environment, and functioning of the family system. A good family positively affects all aspects of family members' development, including their mental health. Many depression-related studies have suggested that family dysfunction is positively correlated with depression to varying degrees [12,13], and numerous studies have confirmed that family functioning is an important predictor of depression onset in adolescents [14,15]. Psychological resilience refers to an individual's ability to "bounce back" from the stress and setbacks in life; such resilience emphasizes an individual's ability to adapt well to adversities, traumas, and threatening situations [16]. As a positive quality, psychological resilience may cushion against the negative impacts brought about by risk factors [17]. The framework of resilience inaction suggests that adolescents' access to good external assets (family functioning) contributes to the development of their internal assets (e.g., psychological resilience), and that good family functioning may promote the development of psychological resilience [17]. Conversely, adolescents who are chronically exposed to poor family environments may have impaired psychological resilience, making them unable to cope with negative external influences and leading to depression. Indeed, several studies have documented that psychological resilience significantly and negatively predicts the onset of depression in adolescents [18,19]. However, little is known about whether it mediates the relationship between family functioning and adolescent depression.

In order to address the gaps mentioned above, a cross-sectional survey was carried out among high school students in Wuhan. The first aim of the current study is to understand the prevalence of depression among adolescents during the normalization stage of the COVID-19 epidemic. The second purpose is to explore the independent effect of external resources (family functioning) and internal resources (psychological resilience) on depression. We speculated that good family functioning and strong psychological resilience could significantly decrease the risk of adolescent depression (Hypothesis 1). The last purpose is to examine the indirect effect of family functioning on depression. We anticipated that psychological resilience mediates the link between family functioning and adolescent depression (Hypothesis 2).

## Materials and methods

### Study design and participants

From September to December 2022, a cross-sectional study was performed by Wuhan Mental Health Center. Participants were recruited from a secondary vocational school (grades 10–12) in Wuhan by using a cluster sampling method. Inclusion criteria were: (1) 15~18 years old; and (2) could be individually matched with their primary caregivers. Exclusion criteria is: (1) nationality was not Chinese.

The staff from Wuhan Mental Health Center, who have received unified training, conducted on-site paper questionnaire surveys among students. Students were asked to complete the questionnaires independently onsite. In cases where respondents do not answer seriously or lack the ability to answer, the investigators will take the initiative to communicate directly with them to boost their motivation to answer and the response rate.

The study protocol was reviewed and approved by the Ethics Committee of Wuhan Mental Health Center (KY2022.11.01). Before the formal survey, informed consent forms which fully disclosed the investigating institution, purpose, procedures, potential benefits and risks of this survey were distributed to students and their legal guardians. All

 

students must obtain the consent of their legal guardians before they can participate in the survey. The survey was anonymized and information provided by the participants will not disclose their identities.

A total of 2,500 questionnaires were distributed. Subjects with missing important survey information, a history of neurological, psychiatric, or those who had experienced major stressful events (such as the sudden death of a close relative, severe violent abuse, suffering from severe diseases, etc.) in the past 12 months were excluded. Finally, 2,410 valid responses were included in the analysis, with a validity rate of 96.4%.

## Measure

**Family functioning.** The Adaptation, Partnership, Growth, Affection, and Resolving Caring Index Questionnaire (APGAR), designed by Smilkstein was used to assess participants' family functionality. It consists of five items: family adaptation, partnership, growth, affection, and resolve. The scale was positively scored, with "rarely," "sometimes," and "often" scoring 0, 1, and 2, respectively. Five entries were added to create a total score, with higher scores representing higher levels of family care and better family functionality. The Cronbach's α coefficient was 0.897 in the present study.

**Psychological resilience.** Psychological resilience was measured using a brief version of the Connor-Davidson Resilience Scale (CD-RISC-10). The scale consists of ten items, each scored on a 5-point scale, with "never," "rarely," "sometimes," "often" and "almost always" scoring 0, 1, 2, 3, and 4, respectively. Higher total scores indicate better psychological resilience. The Cronbach's α coefficient in the current survey was 0.944.

**Depression symptoms.** Depression was assessed using the Patient Health Questionnaire-9 items (PHQ-9), which is a commonly used depression screening tool worldwide. The scale consists of nine items, each scored on a 4-point scale, with "not at all," "a few days," "more than half the days," and "almost every day" scoring 0, 1, 2, and 3 points, respectively. Higher total scores indicate more severe depression. Those with a total score greater than 4 points were determined to have depressive symptoms. The Cronbach's α coefficient in the current survey was 0.922.

**Statistical analysis.** Continuous variables that follow a normal distribution are described as the mean ± standard deviation (M ± SD). For those that do not follow a normal distribution, the median and interquartile range (Median $(Q_{25}, Q_{75})$) is used to describe their distribution characteristic. Categorical variables were described as frequencies (percentages). The $\chi^2$ test was performed to examine the statistical differences in the depression detection rates among two or more demographic characteristics. Multivariate logistic regression analysis was performed to examine the independent effects of family functioning and psychological resilience on depression symptoms, odds ratio (OR) is the primary effect indicator. A structural equation model (SEM) was constructed using the PROCESS macro program to examine the mediating effect of psychological resilience on the relationship between family functioning and depression. In SEM, the root mean square error of approximation (RMSEA) value is < 0.08, and comparative fit index (CFI) and Tucker Lewis index (TLI) values are > 0.90, indicating acceptable models. All analyses were conducted by IBM SPSS 22. All statistical tests were two-sided and P < 0.05 was considered statistically significant.

## Results

### Descriptive data and status of depression symptoms

As presented in Table 1, a total of 2,410 high school students were included in the analysis (1,097 males and 1,313 females), with a mean age of 16.95 ± 0.81 years. Of these, 853, 822, and 735 students were first-, second-, and third-year students, respectively. Depression was detected in 826 students, with a detection rate of 35.77%. Specifically, depressive symptoms were detected in 343 (31.27%) male students and 519 (39.53%) female students. The scores for psychological resilience, family functioning, and depression symptoms were 32(29–39), 5(4–8), and 2(0–7) points, respectively.

A sex difference was observed in the detection of depression, with females having a significantly higher detection rate than males ($\chi^2 = 17.752$, $p < 0.001$). The detection rate was significantly higher in the smoking population than in the non-smoking population ($\chi^2 = 41.185$, $p < 0.001$) and in the drinking population than in the non-drinking population

**Table 1. Distribution of the levels of depression, family functioning and psychological resilience across demographic characteristics and prevalence of depression symptoms.**

| Variables | N (%) | Scores of key scales | | | Depression symptoms | | |
|---|---|---|---|---|---|---|---|
| | | PHQ-9 | APGAR | CD-RISC-10 | N (%) | χ² | p |
| **Total population** | 2410 | 2 (0-7) | 5 (4-8) | 32 (29-39) | 862(35.77) | | |
| **Gender** | | | | | | 17.752 | <0.001 |
| Male | 1097(45.52) | 2 (0-6) | 6 (4-8) | 35 (30-41) | 343(31.27) | | |
| Female | 1313(54.48) | 3 (0-8) | 5 (3-8) | 31 (28-37) | 519(39.53) | | |
| **The only child** | | | | | | 0.032 | 0.859 |
| Yes | 1250(51.87) | 2 (0-7) | 6 (4-9) | 33 (29-40) | 445(35.60) | | |
| No | 1160(48.13) | 2 (0-7) | 5 (3-7) | 32 (29-39) | 417(35.95) | | |
| **Grade** | | | | | | 0.138 | 0.933 |
| Senior one | 853(35.39) | 2 (0-7) | 6 (4-8) | 33 (29-40) | 308(36.11) | | |
| Senior two | 822(34.11) | 2 (0-7) | 5 (3-8) | 31 (29-39) | 295(35.89) | | |
| Senior three | 735(31.50) | 2 (0-7) | 5 (4-8) | 32 (30-40) | 259(35.24) | | |
| **Educational level of father** | | | | | | 2.983 | 0.394 |
| Illiterate and Primary school | 392(16.27) | 2 (0-7) | 5 (3-8) | 32 (29-39) | 152(38.78) | | |
| Junior high school | 776(32.20) | 2 (0-7) | 5 (4-8) | 33 (29-38) | 270(34.79) | | |
| Senior high school | 758(31.45) | 2 (0-7) | 6 (4-8) | 32 (29-40) | 277(36.54) | | |
| University/College or above | 484(20.08) | 2 (0-7) | 6 (4-9) | 33 (29-40) | 163(33.68) | | |
| **Educational level of mother** | | | | | | 1.429 | 0.699 |
| Illiterate and Primary school | 472(19.59) | 2 (0-6) | 5 (3-7) | 32 (29-39) | 163(34.53) | | |
| Junior high school | 766(31.78) | 2 (0-7) | 5 (3-8) | 32 (29-38) | 286(37.34) | | |
| Senior high school | 707(29.33) | 2 (0-7) | 6 (4-8) | 33 (29-40) | 246(34.79) | | |
| University/College or above | 465(19.30) | 2 (0-7) | 6 (4-9) | 33 (29-40) | 167(35.91) | | |
| **Smoking** | | | | | | 41.185 | <0.001 |
| Yes | 204(8.46) | 6(0-12) | 5 (2-6) | 30 (24-37) | 115(56.37) | | |
| No | 2206(91.54) | 2 (0-7) | 6 (4-8) | 33 (29-40) | 747(33.86) | | |
| **Drinking** | | | | | | 78.274 | <0.001 |
| Yes | 723(30.00) | 4(0-9) | 5 (2-7) | 31 (27-38) | 354(48.96) | | |
| No | 1687(70.00) | 2(0-6) | 6 (4-8) | 33 (30-40) | 508(30.11) | | |
| **Family economic situation** | | | | | | 86.649 | <0.001 |
| Very poor | 64(2.66) | 7(2.5-15.5) | 4 (1.5-6.5) | 27.5 (20-38) | 41(64.06) | | |
| Worse | 224(9.29) | 5 (1-9.5) | 4 (1-6) | 30 (25-36) | 126(56.25) | | |
| General | 1639(68.01) | 2 (0-7) | 5 (4-8) | 32 (29-39) | 564(34.41) | | |
| Good | 402(16.68) | 2 (0-6) | 6 (5-9) | 35 (30-41) | 119(29.60) | | |
| Very good | 81(3.36) | 0(0-3) | 9 (6-10) | 40 (30-47) | 12(14.81) | | |
| **Prior mental health conditions** | | | | | | 71.703 | <0.001 |
| Not bad | 1933(80.21) | 2(0-6) | 6 (4-8) | 33 (30-40) | 612(31.66) | | |
| Bad | 477(19.79) | 5(1-10) | 5 (2-7) | 30 (25-36) | 250(52.41) | | |

($x^2$=78.274, $p<0.001$). Depression detection rate among different family economic situations ($x^2$=86.649, $p<0.001$) and prior mental health conditions ($x^2$=71.703, $p<0.001$) was significantly different. No significant difference was observed in the detection rate among subjects with different statuses in terms of whether they were only children, grade level, or parental educational level.

## Logistic regression analysis

Table 2 presents the results of the effects of family functioning and psychological resilience on depression symptoms. Both family functioning and psychological resilience were significantly and negatively associated with the prevalence of depression, with OR=0.800 (95%CI: 0.772–0.830) and OR=0.950 (95%CI: 0.938–0.962) respectively.

Female students (OR=1.315, 95%CI: 1.081–1.601), those who drink alcohol (OR=1.603, 95%CI: 1.287–1.997), and those with poor prior psychological conditions (OR=1.698, 95%CI: 1.346–2.141) have a higher risk of suffering from depression. Compared with students from families with extremely poor economic conditions, students with general (OR=0.457, 95%CI: 0.253–0.827), good (OR=0.477, 95%CI: 0.254–0.896), or excellent economic conditions (OR=0.230, 95%CI: 0.093–0.566) all have a lower prevalence of depression.

## Mediation analysis

As shown in Fig 1, statistically significant covariates were included in the SEM analysis, there were direct effects of family functioning ($\beta$ = -0.575, 95%CI: -0.650,0.500) and psychological resilience ($\beta$ = -0.131, 95%CI: -0.157, -0.105) on adolescent depression symptoms.

As reported in Table 3, the total effect of family functioning on adolescent depression symptoms was -0.710 (95%CI: -0.782, -0.638). Psychological resilience partially mediated the effect of family functioning on depression, with the mediating effect of -0.135 (95% CI: -0.173, -0.099), accounting for 19.72% of the total effect.

**Table 2. Multivariate logistic regression of depression symptoms.**

| Variables | Crude model | | Adjusted model | |
|---|---|---|---|---|
| | OR (95%CI) | P | OR (95%CI) | P |
| **APGAR-score** | 0.778(0.752-0.806) | <0.001 | 0.800(0.772-0.830) | <0.001 |
| **CD-RISC-10 score** | 0.943(0.931-0.955) | <0.001 | 0.950(0.938-0.962) | <0.001 |
| **Gender** | | | | |
| Male | – | – | | |
| Female | – | – | 1.315(1.081-1.601) | 0.006 |
| **Drinking** | | | | |
| No | – | – | | |
| Yes | – | – | 1.603(1.287-1.997) | <0.001 |
| **Smoking** | | | | |
| No | – | – | | |
| Yes | – | – | 1.376(0.959-1.975) | 0.083 |
| **Prior mental health conditions** | | | | |
| Not bad | – | – | | |
| Bad | – | – | 1.698(1.346-2.141) | <0.001 |
| **Family economic situation** | | | | |
| Very poor | – | – | | |
| Worse | – | – | 0.729(0.381-1.397) | 0.341 |
| General | – | – | 0.457(0.253-0.827) | 0.010 |
| Good | – | – | 0.477(0.254-0.896) | 0.021 |
| Very good | – | – | 0.230(0.093-0.566) | 0.001 |

Crude model: Unadjusted

Crude model: Adjusted for sex, smoking, drinking, prior mental health conditions, family economic situation

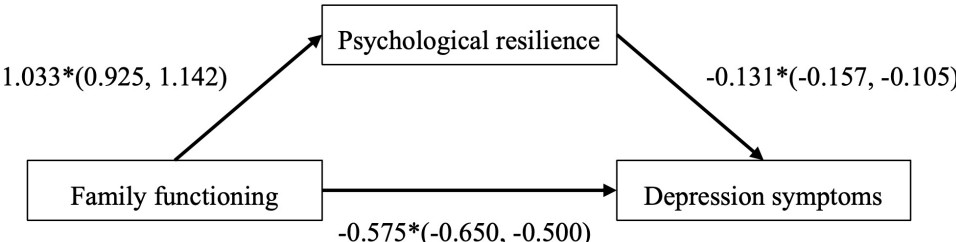

**Fig 1. The effects of family functioning on depression, via psychological resilience.** The overall fit of the model was acceptable: CFI = 0.956, TLI = 0.947, RMSEA = 0.040. * p < 0.001, covariates included sex, smoking status, alcohol consumption, family economic situation and prior mental health conditions.

**Table 3. Direct and indirect effects of family functioning on depression.**

| Effects | β (95%CI) | SE | Proportion of total effect |
|---|---|---|---|
| **Total effects** | -0.710 (-0.782, -0.638) | 0.037 | |
| **Direct effects** | -0.575 (-0.655, -0.505) | 0.038 | 80.28% |
| **Indirect effects** | -0.135 (-0.173, -0.099) | 0.019 | 19.72% |

## Discussion

In the current study, we first explored the situation of adolescent depression among high school students in Wuhan during the normalization stage of the COVID-19 epidemic and investigated the direct and indirect effects of family functioning on adolescent depression. There are some significant findings. First, during the normalization stage of the COVID-19 epidemic, the detection rate of depression among high school students in Wuhan has decreased compared with that during the outbreak stage of the epidemic, but it still remains at a relatively high level. Second, family functioning was not only directly associated with depression but also influenced it through the mediating effect of psychological resilience. These findings enhance our understanding of the development of adolescent depression, which is beneficial for the individual, family, school and social levels to develop targeted adolescent depression prevention measures.

### Detection of depression among high school students

The results indicated that during the regular prevention and control of the COVID-19 pandemic period, 862 (35.77%) of the 2,410 high school students in Wuhan had depressive symptoms. Data from a large-sample survey across China revealed that the depression detection rate of high school students nationwide during the pandemic in 2020 was 43.7% [20]. Another study on the mental health and influencing factors of adolescents in Jingzhou, a city also located in Hubei Province, established that during the regular pandemic prevention and control period, the depression detection rate for adolescent students was 25.8% [21]. A comparison of the results of these studies revealed that, compared with the outbreak period of the pandemic, the detection rate of depression among adolescents in Wuhan was slightly lower during the regular prevention and control of COVID-19. This may be because the pandemic was under better control than it was during the emergency phase, thereby allowing for population mobility and social interactions to return to normal, which made adolescents feel better. However, during the regular prevention and control period, the detection rate was slightly higher in Wuhan than in other cities, indicating that the mental health of adolescents in Wuhan was still greatly affected by the pandemic. Further attention is required to help adolescent students alleviate their bad moods so as to better adapt to their academic status and daily lives.

## Impact of family functioning on adolescent depression

Family functioning significantly and negatively predicted adolescent depression, which is consistent with previous findings [15] and supports our hypothesis. The results of a meta-analysis study found that the association between family dysfunction and depression yielded an OR of 3.72 (95% CI:2.70 to 5.12) [22], which also indirectly supports the findings of our study. Family is an important environment for the growth of adolescents, and a good family atmosphere and parent–child relationship can make adolescents feel love from their parents and prevent the formation of sensitive and withdrawn personalities, thus avoiding depression caused by a lack of sense of security and trust in interpersonal interactions. In effectively functioning families, where close intimacy and cohesion exist among family members, parents have relatively less psychological and behavioral control over their children, while in a relaxed family atmosphere, children tend to have a healthier mindset, are better at expressing emotions, and are able to perceive changes in their moods [23]. Good family functioning can protect adolescents' mental health, and a high degree of family cohesion can provide adolescents with a warm family environment and emotional support, which promotes the formation of favorable parent–child relationships to reduce adolescent depression [24]. A high level of family adaptability may help adolescents cope with the impact of external events, and good family communication may enhance family cohesion and adaptability and help maintain the stability and balance of the family system, thereby reducing the onset of depression. Consequently, parents should respect and understand their children, foster a warm family atmosphere, offer sufficient opportunities for parent-child communication and companionship, and promptly alleviate their children's negative emotions, so as to promote their mental well-being. In this way, even when confronted with major public health events, adolescents can maintain a relatively good state of mental health.

## Mediating role of psychological resilience

The results of this study support the hypothesis that psychological resilience mediates the relationship between family functioning and depression; that is, family functioning can both directly and indirectly influence adolescent depression through psychological resilience. Good family functioning is an important external resource that contributes to the development of adolescents' internal assets (e.g., psychological resilience) [25,26]. Specifically, good family functioning enables adolescents to receive emotional support from their parents [27], which may help them face setbacks and adversity more calmly and further enhance their level of psychological resilience. The present study demonstrated that students with high psychological resilience have a lower risk of suffering from depression, many studies have also presented similar conclusions [28,29]. According to the broadening-and-build theory of positive psychology [30], people with high psychological resilience are better able to broaden their attention, cognition, and action and typically have more positive moods and optimistic attitudes [31], which further provides adolescents with the necessary psychological resources to reduce depression.

Few studies have examined the mediating role of psychological resilience in the relationship between family functioning and depression among adolescent populations. However, studies on certain refugee and diseased populations have confirmed the protective mediating role of psychological resilience in the relationship between family functioning and depression [32,33]. The current evidence suggests that psychological resilience can be enhanced through practice and training [34,35]. Research suggests that interventions should focus on developing assets and resources for at-risk adolescents rather than on more traditional approaches that focus on risk amelioration [16]. Several interventions consider family in efforts to develop assets and resources, such as the Multidimensional Family Prevention project and the Iowa Strengthening Families programs, which are examples of resilient approaches that focus on building positive relationships as a way to prevent negative outcomes. These programs emphasize the importance of family members as resources for the healthy development of adolescents.

In addition, quite a few ways were proposed by researchers to enhance students' psychological resilience apart from those based on family functioning. These include creating virtual reality programs that simulate real-life dilemmas. In

virtual scenarios such as family conflicts, academic pressure, and social setbacks, teenagers need to independently solve problems and make coping decisions. Through repeated experiences, they can familiarize themselves with the stress-coping process and thus improve their psychological resilience [36]. In addition, natural healing spaces can be set up on campus, such as small gardens and green plant corners, equipped with comfortable facilities. Students are encouraged to visit these spaces when they feel stressed or depressed. By getting in touch with nature, they can relax both physically and mentally and restore their psychological energy [37]. Schools can also widely establish various clubs according to students' hobbies, allowing teenagers to study in-depth, create, and showcase their talents in fields they are interested in. During the process of overcoming difficulties, mastering new skills, and gaining recognition from their peers and teachers, students can enhance their self-efficacy and strengthen their psychological resilience [38].

This research suggested that promoting students' psychological resilience can be effective in preventing depression. In light of these findings, it is proposed that psychological resilience enhancing courses should be integrated into the mental health education system of primary and secondary school students as core curricula. Moreover, the assessment of psychological resilience should be incorporated into student mental health evaluations, and special attention should be paid to the students with insufficient psychological resilience.

## Strengths and Limitations

The advantage of this study is that it was conducted in Wuhan, which has a clear locational advantage for investigating the impact of sudden public health events on the mental health of the population. Exploration of the associations between family functioning, psychological resilience, and depression among high school students in Wuhan during the regular prevention and control of the COVID-19 pandemic undeniably serves as guidance for subsequent mental health education and interventions in response to sudden public health events. Second, this study confirmed that psychological resilience mediates the relationship between family functioning and depression among adolescents, which provides strong evidence to support previous studies and is a valid supplement to relevant research on adolescents. Furthermore, more than 2,000 participants were surveyed, and the response rate was remarkably high. Therefore, the study conclusions are considered reliable.

However, this study has several limitations. First, participants were from only one high school, these students cannot fully represent adolescents in Wuhan or broader China, which may limit the generalizability of the findings, larger-scale longitudinal studies from more regions and a wider variety of schools (including vocational high schools and ordinary high schools) should be conducted to further validate our research findings. Second, although some measures have been taken to reduce biases in our survey research, a certain degree of recall and social desirability bias might exist in the present study. Third, given the cross-sectional design of the study, no causal relationship can be established.

## Conclusion

The findings reveal that during the normalization stage of the COVID-19 epidemic, the depression detection rate among high school students in Wuhan remains at a relatively elevated level. Thus, it is of utmost necessity to proactively monitor and address the mental health issues of these students. Family functioning has direct and indirect effects on adolescent depression, and psychological resilience partially mediates the relationship between family functioning and depression. These findings indicate that bolstering psychological resilience can enable Chinese adolescents to mitigate the adverse impacts of poor family functioning on their mental well-being. Moreover, optimizing family functioning and nurturing psychological resilience may be conducive to the prevention of adolescent depression.

## Acknowledgments

The authors are grateful to the cooperation of students from Wuhan finance and trade vocational school.

## Author contributions

**Conceptualization:** Yang Zhou, Shu Yan, Na Lv, Yaofei Xie, Taimin Wu.

**Formal analysis:** Yameng Feng.

**Funding acquisition:** Yang Zhou.

**Visualization:** Cen Gao.

**Writing – original draft:** Yameng Feng.

**Writing – review & editing:** Wenzhen Li, Qingzhou Cheng, Rong Nie, Hongping Zhang, Dajie Chen.

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
