## [Decision Letter · Decision Letter 0]

17 Dec 2024

PONE-D-24-43593The relationship between family function and depression among adolescents in China during the normalization stage of the COVID-19 epidemic:The mediating role of resiliencePLOS ONE

Dear Dr. Feng,

Thank you for submitting your manuscript to PLOS ONE. After careful consideration, we feel that it has merit but does not fully meet PLOS ONE’s publication criteria as it currently stands. Therefore, we invite you to submit a revised version of the manuscript that addresses the points raised during the review process.

Dear Authors,

Two reviewers have considered a number of suggestions to your manuscript.

Also, I give my opinions which must be addressed along with Reviewer responses.

ACADEMIC EDITOR COMMENTS

1. Methods

Strengths:

• The study employs validated scales, enhancing the reliability of the data.

• The large sample size (n=2,410) and high response rate (96.4%) lend statistical power to the findings.

• Structural equation modeling is appropriately utilized for testing mediating effects.

Concerns:

1. Selection Bias: The participants are drawn from a single secondary vocational school in Wuhan. This limits the generalizability of the findings. The authors should clarify if these students are representative of adolescents in Wuhan or broader China.

2. Control of Confounding Variables: While some demographic variables (e.g., sex, smoking, and alcohol use) are controlled, other potentially important factors, such as socioeconomic status and prior mental health conditions, are not addressed.

4. Questionnaire Self-Reporting: Reliance on self-reported data raises concerns about social desirability and recall biases. Discuss it at the end of the manuscript.

2. Results

Strengths:

• The descriptive data is comprehensive, with subgroup analyses by gender and behavioral risk factors.

• The mediating role of resilience is well-documented, with clear statistical support.

Concerns:

1. Ambiguity in Statistical Reporting: The presentation of mediation analysis (Table 4, Figure 1) lacks clarity. The text repeatedly emphasizes indirect and direct effects but fails to sufficiently elaborate on their real-world implications.

2. Interpretative Overreach: The manuscript concludes that family function has a protective effect against depression via resilience, but the findings are correlational. This should be stated more cautiously. Avoid implying causality in discussions of resilience as a mediator

3. Discussion

Strengths:

• The discussion integrates findings with relevant literature and offers practical recommendations for intervention (e.g., resilience training programs).

• The authors highlight the nuanced impact of COVID-19's normalization phase on adolescent mental health, distinguishing it from the emergency phase.

Concerns:

1. The discussion assumes that findings are broadly applicable despite the study's limited geographic scope.

2. Propose novel, context-specific interventions beyond resilience training: While resilience training is widely recognized, the discussion does not provide innovative strategies or context-specific interventions (e.g., leveraging technology for mental health support).

3. Expand on how the findings can inform existing national policies: The paper references national initiatives on adolescent mental health but does not sufficiently engage with how the findings can inform or enhance these programs.

4. Overall Assessment

Decision: Revision Major

Once these concerns are addressed, the manuscript will be significantly strengthened and may be considered for publication in this high-impact journal.

We look forward to receiving your revised manuscript.

Kind regards,

Javier Fagundo-Rivera, PhD

Academic Editor

PLOS ONE

Journal Requirements:

2. Thank you for stating the following financial disclosure: [National Center for Mental Health, China Education Development Foundation, Center for Student services and Development, ministry of education, P.R. China�“Construction and Evaluation of a Collaborative Model for Mental Health Services in Education for Primary and Secondary School Students” (XS24A035)].

3. In this instance it seems there may be acceptable restrictions in place that prevent the public sharing of your minimal data. However, in line with our goal of ensuring long-term data availability to all interested researchers, PLOS’ Data Policy states that authors cannot be the sole named individuals responsible for ensuring data access (http://journals.plos.org/plosone/s/data-availability#loc-acceptable-data-sharing-methods ).

Before we proceed with your manuscript, please also provide non-author contact information (phone/email/hyperlink) for a data access committee, ethics committee, or other institutional body to which data requests may be sent. If no institutional body is available to respond to requests for your minimal data, please consider if there any institutional representatives who did not collaborate in the study, and are not listed as authors on the manuscript, who would be able to hold the data and respond to external requests for data access? If so, please provide their contact information (i.e., email address). Please also provide details on how you will ensure persistent or long-term data storage and availability."

6. Please include captions for your Supporting Information files at the end of your manuscript, and update any in-text citations to match accordingly. Please see our Supporting Information guidelines for more information: http://journals.plos.org/plosone/s/supporting-information .

Additional Editor Comments:

Dear Authors,

Two reviewers have considered a number of suggestions to your manuscript.

Also, I give my opinions which must be addressed along with Reviewer responses.

ACADEMIC EDITOR COMMENTS

1. Methods

Strengths:

• The study employs validated scales, enhancing the reliability of the data.

• The large sample size (n=2,410) and high response rate (96.4%) lend statistical power to the findings.

• Structural equation modeling is appropriately utilized for testing mediating effects.

Concerns:

1. Selection Bias: The participants are drawn from a single secondary vocational school in Wuhan. This limits the generalizability of the findings. The authors should clarify if these students are representative of adolescents in Wuhan or broader China.

2. Control of Confounding Variables: While some demographic variables (e.g., sex, smoking, and alcohol use) are controlled, other potentially important factors, such as socioeconomic status and prior mental health conditions, are not addressed.

4. Questionnaire Self-Reporting: Reliance on self-reported data raises concerns about social desirability and recall biases. Discuss it at the end of the manuscript.

2. Results

Strengths:

• The descriptive data is comprehensive, with subgroup analyses by gender and behavioral risk factors.

• The mediating role of resilience is well-documented, with clear statistical support.

Concerns:

1. Ambiguity in Statistical Reporting: The presentation of mediation analysis (Table 4, Figure 1) lacks clarity. The text repeatedly emphasizes indirect and direct effects but fails to sufficiently elaborate on their real-world implications.

2. Interpretative Overreach: The manuscript concludes that family function has a protective effect against depression via resilience, but the findings are correlational. This should be stated more cautiously. Avoid implying causality in discussions of resilience as a mediator

3. Discussion

Strengths:

• The discussion integrates findings with relevant literature and offers practical recommendations for intervention (e.g., resilience training programs).

• The authors highlight the nuanced impact of COVID-19's normalization phase on adolescent mental health, distinguishing it from the emergency phase.

Concerns:

1. The discussion assumes that findings are broadly applicable despite the study's limited geographic scope.

2. Propose novel, context-specific interventions beyond resilience training: While resilience training is widely recognized, the discussion does not provide innovative strategies or context-specific interventions (e.g., leveraging technology for mental health support).

3. Expand on how the findings can inform existing national policies: The paper references national initiatives on adolescent mental health but does not sufficiently engage with how the findings can inform or enhance these programs.

4. Overall Assessment

Decision: Revision Major

Once these concerns are addressed, the manuscript will be significantly strengthened and may be considered for publication in this high-impact journal.

Reviewers' comments:

Reviewer's Responses to Questions

**Comments to the Author**

1. Is the manuscript technically sound, and do the data support the conclusions?

Reviewer #1: Yes

Reviewer #2: Yes

2. Has the statistical analysis been performed appropriately and rigorously? 

Reviewer #1: Yes

Reviewer #2: No

3. Have the authors made all data underlying the findings in their manuscript fully available?

Reviewer #1: Yes

Reviewer #2: No

4. Is the manuscript presented in an intelligible fashion and written in standard English?

Reviewer #1: Yes

Reviewer #2: Yes

5. Review Comments to the Author

**Reviewer #1: **

Reviewer comment and suggestion

Generally congratulation for the hard work on writing this manuscript. However several issue should be improve.

Adhere on JOURNAL guideline

Work extensively to be clear grammar and typographical errors throughout the document

ABSTRACT

On part of method the authors should explain what type of design and approach used also study population.

I noted line no 28 make it clear.

INTRODUCTION

Introduction well written but I noted to line 107-113 the authors should put it clear in order to remove the confusion to the reader.

METHOD

The authors should give the description of the design also eligibility criteria for participant with sufficient detail to allow replication, including how and when they were actually administered.

How sample size was determined? I noted the authors on Sample size calculations are not written the authors should check and make it clear

It’s a protocol this manuscript?

Nb

On part of methodology there are information are missing the authors should revise and improve them to make this manuscript scientific sound.

RESULT

The author should follow the journals guideline on writing the result

ETHICAL CONSIDERATION

How will you address ethical issue concerning to your study population particularly age 15-18years

LIMITATIONS

The authors should revise and improve are not clear.

REFFERENCES

Several references do not fit the requirements of Vancouver style. Revise and improve them

**Reviewer #2: **

Hello dear authors.

MS Id: PONE-D-24-43593

Title: The relationship between family function and depression among adolescents in China during the normalization stage of the COVID-19 epidemic: The mediating role of resilience

Type: Original Research

Here are my recommendations about the mentioned MS:

Abstract:

• Looks good.

Introduction:

• Adding reference for line 81, 86, and 100.

• State and describe your problem better.

Methodology:

• State your study design.

• How did validity for questionnaires done?

• Please write down the code of ethics you received from the institution you mentioned.

• Dedicate a section in methodology for ethical approval and inform consent.

• Explain the chi square test.

• How did you determine the level of depression symptom

• Explain the structural equation modeling technique more clearly.

• Provide a normality test and explain whether your data is normal or not.

Results:

• I suggest using regression tests for predicting factors.

Discussion:

• There's no need to divide the discussion into sub-sections.

Conclusion:

• Looks fine.

References:

• Looks good.

Figures and tables:

• Provide a table for descriptive data and status of depression symptoms.

Some more issues should be considered necessary for publication:

• Suggestions for future studies also be mentioned.

6. PLOS authors have the option to publish the peer review history of their article (what does this mean? ). If published, this will include your full peer review and any attached files.

**Do you want your identity to be public for this peer review?** For information about this choice, including consent withdrawal, please see our Privacy Policy .

Reviewer #1: No

Reviewer #2: No

---

## [Decision Letter · Decision Letter 1]

12 Feb 2025

PONE-D-24-43593R1The relationship between family function and depression among adolescents in China during the normalization stage of the COVID-19 epidemic:The mediating role of resiliencePLOS ONE

Dear Dr. Chen,

Thank you for submitting your manuscript to PLOS ONE. After careful consideration, we feel that it has merit but does not fully meet PLOS ONE’s publication criteria as it currently stands. Therefore, we invite you to submit a revised version of the manuscript that addresses the points raised during the review process.

**ACADEMIC EDITOR:** Dear Authors,

Thank you for appropriately revising the manuscript.

The two reviewers appreciate your modifications, although they have provided additional comments to further improve the manuscript.

In this regard, my own comments have been properly addressed.

Therefore, I recommend a second round (R2) for **minor revisions** .

We look forward to receiving your revised manuscript.

Kind regards,

Javier Fagundo-Rivera, PhD

Academic Editor

PLOS ONE

Journal Requirements:

Additional Editor Comments:

Dear Authors,

Thank you for appropriately revising the manuscript.

The two reviewers appreciate your modifications, although they have provided additional comments to further improve the manuscript.

In this regard, my own comments have been properly addressed.

Therefore, I recommend a second round (R2) for minor revisions.

Reviewers' comments:

Reviewer's Responses to Questions

**Comments to the Author**

1. If the authors have adequately addressed your comments raised in a previous round of review and you feel that this manuscript is now acceptable for publication, you may indicate that here to bypass the “Comments to the Author” section, enter your conflict of interest statement in the “Confidential to Editor” section, and submit your "Accept" recommendation.

Reviewer #1: All comments have been addressed

Reviewer #2: (No Response)

2. Is the manuscript technically sound, and do the data support the conclusions?

Reviewer #1: Yes

Reviewer #2: Yes

3. Has the statistical analysis been performed appropriately and rigorously? 

Reviewer #1: Yes

Reviewer #2: Yes

4. Have the authors made all data underlying the findings in their manuscript fully available?

Reviewer #1: Yes

Reviewer #2: Yes

5. Is the manuscript presented in an intelligible fashion and written in standard English?

Reviewer #1: Yes

Reviewer #2: Yes

6. Review Comments to the Author

**Reviewer #1: **

Generally congratulations to authors on putting much effort for this manuscript well comment improving however few issues need to make it clear and understandable for the reader.

• Work extensively to be clear grammar and typographical errors throughout the document.

• Also on top of your title authors should write the study design.

• I noted On part of *abstract * (methods) the authors should follow the sequence for starting with approach used then study design, starting date and ending date ,sampling procedure and sample size, the tools and model of analysis used and then statistical analysis improve to make it clear. Also on part of result the authors should remove the one bracket.

*Methods and material*

• I noted line no 146,152,160,161 the authors should clear the statements (not study).

*Discussion*

• I would ask the authors on this study how many objective do you have? Because on this part authors need improving and make it clear.

*Conclusion*

• It better to start with (THE FINDINGS) less than staring with our findings.

**Reviewer #2: **

Hello dear authors.

MS Id: PONE-D-24-43593R1

Title: The relationship between family function and depression among adolescents: The mediating role of resilience

Type: Original Research

Here are my recommendations about the mentioned manuscript:

*Abstract:*

• Looks good.

*Introduction:*

• State the gap better.

*Methodology:*

• How do you controlled the exclusion criteria number 1. Furthermore, no need to mention the criteria number 2.

• Which type of data collection was used?

• How did validity for questionnaires done?

• Dedicate a section in methodology for ethical approval and inform consent.

• Mention the levels of depression, resilience and family function

• I suggest the regression methods.

*Results:*

• Looks good.

*Discussion:*

• There's no need to divide the discussion into sub-sections.

• The three final paragraphs need to be revised and discuss your results with previous study results and facts.

*Conclusion:*

• Need revision and state results clearer.

*References:*

• Looks good.

*Figures and tables:*

• Provide a figure levels of depression, resilience and family functions.

Some more issues should be considered necessary for publication:

• Limitation of the present study need to be mentioned.

• Suggestions for future studies also be mentioned.

• Please provide at least two related strengths for manuscript.

• The manuscript need proofreading by a native speaker.

7. PLOS authors have the option to publish the peer review history of their article (what does this mean? ). If published, this will include your full peer review and any attached files.

**Do you want your identity to be public for this peer review?** For information about this choice, including consent withdrawal, please see our Privacy Policy .

Reviewer #1: **Yes: ** rehema abdallah

Reviewer #2: No

---

## [Author Response · Author response to Decision Letter 2]

28 Mar 2025

Dear Editors and Reviewers:

Thank you for your letter and for the reviewers’ comments concerning our manuscript entitled “The relationship between family function and depression among adolescents in China during the normalization stage of the COVID-19 epidemic: the mediating role of resilience” (ID: PONE-D-24-43593R1). Those comments are all valuable and very helpful for revising and improving our paper, as well as the important guiding significance to our research. We have studied comments carefully and have made corrections which we hope to meet with approval. In the revised manuscript, revised portions are highlighted with red color. The main corrections in the paper and the responds to the reviewers’ comments are as follows.

Reviewer #1:

Generally congratulations to authors on putting much effort for this manuscript well comment improving however few issues need to make it clear and understandable for the reader.

• Work extensively to be clear grammar and typographical errors throughout the document.

Thank you for your suggestions. Our team has already revised the format and layout of the entire paper in accordance with the journal's guidelines.

• Also on top of your title authors should write the study design

Thank you for your suggestions. On top of the title, we have marked the study design as "Original Research" and have highlighted it in red. (The revised parts are marked in red on line 1)

• I noted On part of abstract (methods) the authors should follow the sequence for starting with approach used then study design, starting date and ending date ,sampling procedure and sample size, the tools and model of analysis used and then statistical analysis improve to make it clear. Also on part of result the authors should remove the one bracket.

We are extremely grateful for your valuable suggestions. In the “method” section of the abstract, we have made the corresponding revisions according to your suggestions, which are expressed as “A cross - sectional survey was conducted from September to December 2022 in Wuhan. Using a census methodology, 2,410 students from a secondary vocational school were recruited for the study.” (The revised parts are marked in red on lines 28-30)

Methods and material

• I noted line no 146,152,160,161 the authors should clear the statements (not study).

Thank you for your suggestions. We have revised the expressions regarding the reliability and validity of the three main scales (APGAR, CD-RISC-10, PHQ-9), which are presented as “The Cronbach’s α coefficient was 0.897 in the present study”, “The Cronbach’s α coefficient in the current survey was 0.944”, “The Cronbach’s α coefficient in the current survey was 0.922”. (The revised parts are marked in red on lines 155, 161,168-169)

Discussion

• I would ask the authors on this study how many objective do you have? Because on this part authors need improving and make it clear.

Thank you for your suggestions. In the “Introduction” section of the revised manuscript, the research objectives and significance have been modified to make the expressions more explicit, which is stated as " In order to address the gaps mentioned above, a cross-sectional survey was carried out among high school students in Wuhan. The first aim of the current study is to understand the prevalence of depression among adolescents during the normalization stage of the COVID-19 epidemic. The second purpose is to explore the independent effect of external resources (family functioning) and internal resources (psychological resilience) on depression. We speculated that family functioning and psychological resilience could significantly decrease the risk of adolescent depression (Hypothesis 1). The last purpose is to examine the indirect effect of family functioning on depression. We anticipated that psychological resilience mediates the link between family functioning and adolescent depression (Hypothesis 2)." (The revised parts are marked in red on lines 112-121)

Regarding the content of the “discussion” section, we have improved and supplemented it in the discussion section of the revised manuscript. (The revised parts are marked in red on lines 228-239, 282-284)

Conclusion

• It better to start with (THE FINDINGS) less than staring with our findings.

Thank you for your suggestions. We have made corresponding revisions in the “Conclusion” section and the presentation is as follows: “The findings reveal that during the normalization stage of the COVID-19 epidemic, the depression detection rate among high school students in Wuhan remains at a relatively elevated level. Thus, it is of utmost necessity to proactively monitor and address the mental health issues of these students. Family functioning has direct and indirect effects on adolescent depression, and psychological resilience partially mediates the relationship between family functioning and depression. These findings indicate that bolstering psychological resilience can enable Chinese adolescents to mitigate the adverse impacts of poor family functioning on their mental well-being. Moreover, optimizing family functioning and nurturing psychological resilience may be conducive to the prevention of adolescent depression.” (The revised parts are marked in red on line 361-370)

Reviewer #2:

Abstract:

• Looks good.

Introduction:

• State the gap better.

We sincerely appreciate your pointing out the problems in our manuscript. The “Introduction” section has been supplemented in the revised manuscript, and our research objectives have been further refined.

The supplementary statement is as follows: " Therefore, more research should be conducted to detect the epidemiological characteristics of depression among adolescents in Wuhan during the normalization stage of the COVID-19 epidemic and to explore the preventive approaches to adolescent depression." (The revised parts are marked in red on lines 83-86)

The research objectives are integrated and refined; the expression is as follows: " In order to address the gaps mentioned above, a cross-sectional survey was carried out among high school students in Wuhan. The first aim of the current study is to understand the prevalence of depression among adolescents during the normalization stage of the COVID-19 epidemic. The second purpose is to explore the independent effect of external resources (family functioning) and internal resources (psychological resilience) on depression. We speculated that family functioning and psychological resilience could significantly decrease the risk of adolescent depression (Hypothesis 1). The last purpose is to examine the indirect effect of family functioning on depression. We anticipated that psychological resilience mediates the link between family functioning and adolescent depression (Hypothesis 2)." (The revised parts are marked in red on lines 112-121)

Methodology:

• How do you controlled the exclusion criteria number 1. Furthermore, no need to mention the criteria number 2.

This whole-school census adhered to the principle of voluntariness, students could participate after obtaining consent from their guardians. Therefore, at the level of study design, the exclusion criteria did not include (1) history of neurological, psychiatric, or other serious somatic diseases. I would like to express my sincere apologies for the error in the exclusion criteria described in the previous version of the manuscript. During the data analysis phase, considering the potential interference of the history of neurological, psychiatric and major stressful events on the screening results, participants who answered "yes" to these two questions were excluded.

In the revised manuscript, the exclusion criteria of the study have been revised, and the details of the samples excluded from the analysis have been presented as “Subjects with missing important survey information, a history of neurological, psychiatric, or those who had experienced major stressful events (such as the sudden death of a close relative, severe violent abuse, suffering from severe diseases, etc.) in the past 12 months were excluded.”(The revised parts are marked in red on lines 128, 142-145)

• Which type of data collection was used?

We sincerely appreciate your concern regarding the type of data collection. The data was collected through on-site paper questionnaire surveys from September to November 2022. Through a census approach, a total of 2,500 questionnaires were collected from a secondary vocational school in Wuhan.

The specific steps of data collection were as follows: One week before the investigators conducted the survey, teachers distributed paper copies of the informed consent forms to students, informing them of the survey's purpose, procedures, investigating institutions, and rights protection measures (voluntary participation, confidentiality, etc.). Students and their guardians jointly decided whether to participate in the survey and signed the informed consent forms. On the day of the formal survey, the staff from the Wuhan Mental Health Center distributed the questionnaires in the classrooms and collected the data on the spot.

In the section "study and population" of the revised manuscript, the description has been supplemented and refined. (The revised parts are marked in red on lines 130, 135-141)

• How did validity for questionnaires done?

Dear reviewer�regarding content validity, the scales in the questionnaire are all sourced from validated and well-established scales in authoritative literature of the corresponding fields. These scales have been widely used in numerous previous studies and have been proven through extensive practice to cover the key dimensions related to our research topic.

Regarding construct validity: (1) Construct validity of individual scales: we used Confirmatory Factor Analysis (CFA) to test the construct validity of the scales. Cronbach’s α coefficient of the three main scales (APGAR, CD-RISC-10, PHQ-9) are all greater than 0.9, indicating the scales have good validity. (2) Construct validity of the SEM: the fitting indices of the SEM are as follows: For the overall model, χ²/df = 8.027, CFI = 0.956, TLI = 0.947, and RMSEA = 0.040, the overall fitting of this SEM is excellent. The fitting details of the SEM are provided as supplementary explanations in the caption of Figure 1. (The revised parts are marked in red in figure 1)

• Dedicate a section in methodology for ethical approval and inform consent.

Thank you for your valuable suggestions, In the revised manuscript, the information about ethical approval and inform consent has been presented in a separate paragraph, which stated as " The study protocol was reviewed and approved by the Ethics Committee of Wuhan Mental Health Center (KY2022.11.01). Before the formal survey, informed consent forms which fully disclosed the investigating institution, purpose, procedures, potential benefits and risks of this survey were distributed to students and their legal guardians. All students must obtain the consent of their legal guardians before they can participate in the survey. The survey was anonymized and information provided by the participants will not disclose their identities." (The revised parts are marked in red on lines 135-141)

• Mention the levels of depression, resilience and family function

Thank you for your valuable suggestions. In Table 1 of the revised manuscript, the distribution of the levels of two important variables (family function and psychological resilience) other than depression has been added. (The revised parts are marked in red in Table 1)

• I suggest the regression methods.

Thank you for your valuable suggestions. After discussing with our team members, we highly approve of applying the logistics regression model. Compared with correlation analysis, it can provide more abundant information. Regression can explain the correlation and also calculate the effect (Odds Ratio) of independent variables. We have replaced the correlation analysis with regression analysis in the revised manuscript according to your suggestions, and the results are as follows:

Table 2 presents the results of the effects of family function and psychological resilience on depression symptoms. Both family function and psychological resilience were significantly and negatively associated with the prevalence of depression, with OR =0.800 (95%CI: 0.772-0.830) and OR=0.950 (95%CI: 0.938-0.962) respectively.

Female students�OR=1.315, 95%CI: 1.081-1.601), those who drink alcohol (OR=1.603, 95%CI: 1.287-1.997), and those with poor prior psychological conditions (OR=1.698, 95%CI: 1.346-2.141) have a higher risk of suffering from depression. Compared with students from families with extremely poor economic conditions, students with general�OR=0.457, 95%CI: 0.253-0.827), good�OR=0.477, 95%CI: 0.254-0.896), or excellent economic conditions �OR=0.230, 95%CI: 0.093-0.566) all have a lower prevalence of depression. (The revised parts are marked in red in lines 198-208, Table 2)

Table 2 Multivariate logistic regression of depression symptom

Variables Crude model Adjusted model

OR (95%CI) P OR�95%CI� P

APGAR-score 0.778(0.752-0.806) <0.001 0.800(0.772-0.830) <0.001

CD-RISC-10 score 0.943(0.931-0.955) <0.001 0.950(0.938-0.962) <0.001

Gender

Male - -

Female - - 1.315(1.081-1.601) 0.006

Drinking

No - -

Yes - - 1.603(1.287-1.997) <0.001

Smoking

No - -

Yes - - 1.376(0.959-1.975) 0.083

Prior mental health conditions

Not bad - -

Bad - - 1.698(1.346-2.141) <0.001

Family economic situation

Very poor - -

Worse - - 0.729(0.381-1.397) 0.341

General - - 0.457(0.253-0.827) 0.010

Good - - 0.477(0.254-0.896) 0.021

Very good - - 0.230(0.093-0.566) 0.001

Crude model: Unadjusted

Crude model: Adjusted for sex, smoking, drinking, prior mental health conditions, family economic situation

Besides the results section, we have also made corresponding revisions in the “abstract” and “statistical analysis” section (The revised parts are marked in red in lines 30-32, 35-38, 176-178).

Results:

• Looks good.

Discussion:

• There's no need to divide the discussion into sub-sections.

We sincerely appreciate the suggestions you provided, which are truly valuable. However, we were unable to implement them. This is not in any way a reflection of the practicality of your ideas. Instead, our team decided to take a different approach in the discussion section. We have added comprehensive introductory content regarding the research content, results, and significance, which is now presented in the first paragraph of the discussion section. The specific details are as follows: “In the current study, we first explored the situation of adolescent depression among high school students in Wuhan during the normalization stage of the COVID-19 epidemic and investigated the direct and indirect effects of family functioning on adolescent depression. There are some significant findings. First, during the normalization stage of the COVID-19 epidemic, the detection rate of depression among high school students in Wuhan has decreased compared with that during the outbreak stage of the epidemic, but it still remains at a relatively high level. Second, family functioning was not only directly associated with depression but also influenced it through the mediating effect of psychological resilience. These findings enhance our understanding of the development of adolescent depression, which is beneficial for the individual, family, school and social levels to develop targeted adolescent depression prevention measures.” (The revised parts are marked in red on line 228-239)

After adding this part of the content, we believe that discussing in sub-sections based on the main research findings can make the structure of the discussion section much clearer.

• The three final paragraphs need to be revised and discuss your results with previous study results and facts.

Thank you for your valuable suggestions. In the discussion section, we have additionally cited three references. Based on these, we have conducted a comparative analysis between our research findings and those of previous studies. The presentation is as follows: “The results of a meta-analysis

---

## [Editor Report · Decision Letter 2]

31 Mar 2025

The relationship between family functioning and depression among adolescents in China during the normalization stage of the COVID-19 epidemic:The mediating role of resilience

PONE-D-24-43593R2

Dear Dr. Chen,

We’re pleased to inform you that your manuscript has been judged scientifically suitable for publication and will be formally accepted for publication once it meets all outstanding technical requirements.

Kind regards,

Javier Fagundo-Rivera, PhD

Academic Editor

PLOS ONE

**Additional Editor Comments:**

Dear Authors,

All reviewers' comments have been addressed.

The manuscript can now be accepted for publication in Plos One.

---

## [Editor Report · Acceptance letter]

PONE-D-24-43593R2

PLOS ONE

Dear Dr. Chen,

I'm pleased to inform you that your manuscript has been deemed suitable for publication in PLOS ONE. Congratulations! Your manuscript is now being handed over to our production team.

Kind regards,

on behalf of

Dr. Javier Fagundo-Rivera

Academic Editor

PLOS ONE